# Cyclosporin A as an Add-On Therapy to a Corticosteroid-Based Background Treatment in Patients with COVID-19: A Multicenter, Randomized Clinical Trial

**DOI:** 10.3390/jcm13175242

**Published:** 2024-09-04

**Authors:** Lucía Llanos Jiménez, Beatriz Alvarez-Alvarez, Eva Fonseca Aizpuru, Germán Peces-Barba, Gloria Pindao Quesada, Mª Jesús Rodríguez Nieto, Francisco J. Ruiz-Hornillos, Luis Seijo Maceiras, Ignacio Robles Barrena, Alvaro Mena-de-Cea, Héctor Meijide-Míguez, Olga Sánchez-Pernaute

**Affiliations:** 1Fundación Jiménez Díaz (FJD) University Hospital, FJD Health Research Institute, Universidad Autónoma de Madrid (IIS-FJD, UAM), 28040 Madrid, Spain; balvarez@quironsalud.es (B.A.-A.); gpeces@quironsalud.es (G.P.-B.); mjrodriguezn@fjd.es (M.J.R.N.); osanchez@fjd.es (O.S.-P.); 2Cabueñes Hospital, 33394 Asturias, Spain; evamfonseca@yahoo.es; 3Villalba General University Hospital, FJD Health Research Institute, Universidad Autónoma de Madrid (IIS-FJD, UAM), 28400 Madrid, Spain; gloria.pindao@hgvillalba.es; 4Infanta Elena University Hospital, FJD Health Research Institute, Universidad Autónoma de Madrid (IIS-FJD, UAM), 28342 Madrid, Spain; javier.ruiz@quironsalud.es; 5Faculty of Medicine, Universidad Francisco de Vitoria, 28223 Madrid, Spain; 6Clínica Universitaria de Navarra (CUN), 28027 Madrid, Spain; lseijo@unav.es; 7Rey Juan Carlos University Hospital (HURJC), FJD Health Research Institute, Universidad Autónoma de Madrid (IIS-FJD, UAM), 28933 Madrid, Spain; iroblesb@hospitalreyjuancarlos.es; 8Internal Medicine Department, A Coruña University Hospital Complex, 15006 A Coruña, Spain; alvaro.mena.de.cea@sergas.es; 9Quironsalud Hospital A Coruña, 15009 A Coruña, Spain; hector.meijide@quironsalud.es

**Keywords:** COVID-19 pneumonia, hyperinflammation, cyclosporin A, randomized controlled trial

## Abstract

**Background**: In susceptible hosts, SARS-CoV2-induced hyperinflammation accounts for an increased mortality. The search of adjuvant immunomodulatory therapies has been ongoing ever since the pandemic outbreak. Aim: Our purpose was to evaluate the efficacy of cyclosporin A (CsA) as an add-on therapy to the standard of care (SoC) in patients with severe COVID-19 pneumonia. **Methods**: We conducted a randomized clinical trial in patients admitted to eight Spanish tertiary hospitals. Patients were stratified into two severity categories and randomized in a 1:1 ratio to receive a corticosteroid-based standard therapy with or without CsA. The primary endpoint was FiO2 recovery by Day 12 without relapses. **Results**: 109 patients were included and randomized, and 98 of them considered for the mITT population (51 assigned to the CsA + SoC group and 47 to the SoC group). A total of 35 (68.6%) patients from the CsA + SoC group and 32 (71.1%) patients from the SoC group reached the primary endpoint in the mITT analysis. No differences were found after stratification into age groups, in the severity level at admission, or in a combination of both. Overall, the time to FiO2 normalization was 7.4 days vs. 7.9 days in the experimental and control groups, respectively. Global mortality was 8.2%. Severe adverse events were uncommon and equally distributed between arms. **Conclusion**: The addition of CsA did not show differences over a corticosteroid-based treatment in the clinical course of the included patients. A better identification of candidates who will benefit from receiving immunomodulatory drugs is necessary in future studies.

## 1. Introduction

Already from the early months of the COVID-19 pandemic, it became apparent that the new coronavirus was able to trigger a hyperinflammatory response driving multi-systemic involvement along with respiratory distress [1,2]. In this context, there was a need to find immunomodulatory strategies that are able to dampen lung and systemic inflammation but, at the same time, are also sufficiently safe for critically ill infected patients [3]. On April 2020, the Spanish Ministry of Health encouraged the development of pragmatic clinical trials (CT) aimed at reducing mortality, as well as alleviating hospital and intensive care resource saturation. In a short time frame, we set a randomized CT—the results of which are presented in this article—to assess the efficacy of cyclosporin A (CsA) as an add-on therapy to the standard of care (SoC) in inpatients with COVID-19 pneumonia requiring oxygen supplementation.

Considering that patients with COVID-19-associated hyperinflammation showed features of macrophage activation syndrome [2,4], we postulated that SARS-CoV2 was able to infect myeloid cells, hijacking cell machinery and hindering antiviral responses [5]. This pathogenic pathway, which is shared by different RNA viruses, can result in inflammation/autoimmunity, thereby providing a rationale for the use of immunosuppressants [6,7]. In addition, the involvement of RIG-I sensors and the mitochondrial antiviral signaling (MAVS) protein in betacoronaviridae recognition pointed to mitochondrial dysfunction as a key event in COVID-19 hyperinflammation [8,9,10].

Along with its cornerstone place in the prevention of graft rejection, CsA is widely used for the treatment of autoimmune and inflammatory disorders. Notwithstanding, the fungus derivate has unique characteristics not limited to its tolerogenic profile. In particular, the CsA binding of cyclophilins provides cytoprotection during hypoxia/reperfusion injury and upon the incidence of stressors jeopardizing cellular metabolic processes [11,12]. Moreover, the inhibition of cyclophilins has been shown to account for an effective counterattack strategy against RNA viruses, including coronaviridae [13]. Due to the profound lymphocyte depletion that characterizes the acute phase of COVID-19, both of the antiviral properties of CsA, as well as its good safety profile in patients with HIV, were determinants for our decision to launch this trial [14,15]. The main purpose of this study was to evaluate the efficacy and safety of CsA as add-on therapy to SoC in improving the severity in inpatients with COVID-19 pneumonia and respiratory failure.

## 2. Materials and Methods

### 2.1. Study Design

A national, multicentric, open-label, low-intervention, controlled, and randomized study was designed. The study protocol was approved by the Ethics Committee and Spanish Medicines Agency (AEMPS), and it was prospectively registered before recruitment started (NCT04392531). All subjects provided informed consent before entering the trial. (Amendments to the protocol after trial start are available in Appendix A, and the last current version is available in Spanish in Appendix A.)

### 2.2. Study Population

This study was performed in 8 university hospitals across Spain. Potential eligible participants were adults of both sexes that fulfilled the admission criteria because of COVID-19 pneumonia. Detailed inclusion and exclusion criteria are included in Appendix A

As there were no published studies available at the time of the study design to estimate the CsA effect, the sample size was calculated by hypothesizing that adding CsA to the SoC would increase the proportion of patients normalizing FiO2 values by Day 12 from 70% to 90%. In accepting an alpha risk of 0.05 and a beta risk of 0.2 in a bilateral contrast, 60 subjects were needed in each group to detect, as statistically significant, the predefined difference.

### 2.3. Randomization and Study Treatment

Patients fulfilling the eligibility criteria were randomly assigned in a 1:1 ratio to either the CsA+SoC or SoC group and stratified by severity level on the day of inclusion. This severity classification was based on the British Guideline for oxygen use [16] and an adaptation from a fraction of the inspired oxygen (FiO2)-based risk classification for emergency triage (see Appendix A). For stratification purposes, patients were classified in lower (FiO2 requirements < 60%) and higher (FiO2 requirements ≥ 60%) severity levels. Patients of >80 years were stratified at a higher severity level regardless of oxygen requirements. Randomization was centrally performed using the “blocrank” R package in the Coordinating Hospital Statistics Department. No blinding procedures were performed.

### 2.4. Study Treatment

Patients assigned to the CsA+SoC group started treatment with CsA within 24 h from inclusion according to a dosing schedule, which is described in detail in Appendix A. The duration of treatment was 2 and 4 weeks from inclusion for patients stratified as non-severe and severe, respectively.

### 2.5. Study Procedures

At the screening visit, the demographics, relevant medical history, COVID-19-related signs and symptoms, and CURB-65 [17] scores were registered. FiO2 and the level of supplemental oxygen requirements were registered daily. In addition, physical examination, vital signs, CsA dosing and compliance, concomitant treatments, complications, and adverse events were registered throughout the trial. Routine laboratory parameters including C-reactive protein (CRP), lactate dehydrogenase (LDH), ferritin, creatine phosphokinase (CPK), troponin I, and D-dimer were monitored every 48 h during admission. Urinalysis (including creatinine and Na), interleukin (IL)-6 levels, lymphocyte subpopulations, and CsA plasma concentrations were assessed weekly. Anti SARS-CoV-2 IgG and IgM quantification, as well as SARS-CoV-2 detection, with PCR techniques were conducted at baseline, the 8th day, and at the 15th day of hospitalization, as well as in the end of study (EOS) visit. Chest X ray findings were registered at baseline and in the EOS visit. After discharge, patients were followed by a phone call every 2 days during the first week and every 4 days until the EOS visit to register patient general health status, CsA compliance when applicable, concomitant medications, complications, and adverse events. EOS was performed 4 and 6 weeks from admission depending on the severity level reached during hospital stay (lower vs. higher), respectively.

### 2.6. Study Outcomes

According to the study objectives, the primary outcome was the proportion of patients without oxygen support (or who had returned to baseline FiO2 in case of patients with previous oxygen therapy) at Day 12 without relapse during follow-up. Secondary outcomes included (a) deaths, ICU and hospital stays, and FiO2 change; (b) evolution of blood pressure (BP), plasma creatinine, and total lymphocyte and CD4 counts; (c) adverse events; (d) changes in viral load and seroconversion parameters; (e) impact of CsA on the reduction in ferritin, LDH, and CRP levels from baseline, as well as the peak levels of CPK, D-dimer, and IL6 levels; and (f) patient global assessment at the EOS.

### 2.7. Statistical Considerations

For both efficacy and safety assessments, a modified intention-to-treat (mITT) analysis was performed including all randomized patients who had received at least one dose of the study medication, and a per protocol (PP) analysis was conducted including the patients who had completed the study. A descriptive analysis of the population was performed. The normality of the variables was tested using the Kolmogorov–Smirnov test. To compare the differences between treatment groups, Pearson’s Chi-square, Student’s t-test, or the Mann–Whitney test were used depending on the type of variable. Kaplan–Meier survival curves were estimated and compared with the Mantel–Cox test. In addition, Cox regression models were fitted to estimate the hazard ratio together with its confidence interval. An intermediate analysis for the primary outcome was foreseen and performed when 40% of the study population had reached the 8th day of hospitalization. Subgroup analyses by age and severity level were also performed. The statistical package SAS^®^ 9.3 (SAS Institute Inc., Cary, NC, USA) was used to carry out the analysis.

## 3. Results

### 3.1. Setting and Study Population

AEMPS approval was received on the 9th of April 2020. A total of 109 patients were included between April 2020 and April 2021 (91% of the preplanned sample size). The last center close-out visit was performed on December 2021. All included patients were randomized, and 98 were considered for the mITT analysis. Of them 47 were allocated in the SoC group, and 51 in the CsA + SoC group (see Figure 1). The majority of the patients were stratified into the “lower severity” stratum (43/47 and 46/51, respectively). A total of 2 patients withdrew consent before reaching Day 12 so they could not be included in the mITT population for the primary efficacy outcome analysis. Table 1 shows the participant baseline characteristics regarding host-dependent features and process-related risk factors. No statistically significant differences between groups were found in any of them.

### 3.2. Study Intervention

Both study groups received similar background treatment according to the SoC protocol elaborated and periodically reviewed by a multidisciplinary committee at Coordinating Hospital (detailed in Appendix A).

In the participants assigned to the CsA+SoC group, CsA was administered for 15.1 ± 7.5 days, at an average dose of 177 mg/day per patient, and drawing a mean cumulative dose of 2686.3 ± 1527 mg [95% CI 2256.8, 3115.8].

With regard to immunomodulating agents, methylprednisolone pulses were administered to 55 participants (25 from the SoC group and 30 from the CsA+SoC group). In addition, 29 participants (15 in the SoC group and 14 in the CsA+SoC group) received 0.6 to 1 mg/kg/day of methylprednisolone or equivalent, whereas the administration of rescue medication was performed with 400 mg of intravenous tocilizumab in 10 subjects (6 from the SoC group and 4 from the CsA+SoC group).

### 3.3. Primary Outcome

As shown in Table 2, no significant differences in primary outcomes were found between groups for either of the analysis populations. A total of 32 (71.1%) patients from the SoC group and 35 (68.6%) from the CsA+SoC group reached the primary endpoint in the mITT analysis, while proportions drawn in the PP analysis were 76.3% vs. 75.6%, respectively. No significant differences were found in the subgroup analysis by age (>65 years) or severity level.

Only 9 participants (6 in the PP analysis) were included in the higher severity category at recruitment. Of them, 1 patient in each treatment arm (25% in the SoC group vs. 20% in the CsA+SoC group) achieved the primary endpoint. Intermediate analysis showed no statistically significant difference between the groups for primary outcomes.

### 3.4. Secondary Outcomes

Results regarding mortality, length of hospital stay, and FiO2 evolution are listed in Table 3. There were eight deaths during hospitalization, four in each of the treatment arms, and they were all attributable to a progression of respiratory failure and critical illness-related complications. Hospital length of stay was slightly higher in the CsA+SoC group, particularly in those patients stratified as lower-severity and in patients above 65 years of age. A total of 15 patients were admitted to the ICU during the episode (5 of them allocated to the SoC group and 10 to CsA+SoC group), of whom only 1 patient was found to meet the primary endpoint. No significant differences were observed in any of these comparisons.

Figure 2 shows the time to FiO2 normalization in both arms (A) and the mean FiO2 oxygen requirements during the 12 first days of this study in the ward-admitted population (B). Post hoc analyses exploring daily FiO2 requirements were performed in patient subgroups according to the FiO2 at enrolment and in the age subgroups (see Appendix A). In these graphs, the therapeutic arms parted from Day 6 of the trial in the admitted patients starting with FiO2 < 35% and in those aged ≥ 65. However, the size of these comparisons was low and none of them yielded significant differences.

### 3.5. Safety Measures

Systolic blood pressure (SBP) showed a tendency to drop during hospitalization in patients enrolled in the SoC group. Conversely, patients in the CsA+SoC group showed stable or increasing SBP values, with the difference between arms becoming significant at Day 16 (*p* 0.031). On the other hand, no differences were observed in the diastolic BP. Plasma creatinine, which was the principal analytical safety measure, remained stable during admission.

A total of 130 adverse events occurred, most of them of mild intensity (86.2%), with 46 in the SoC group and 84 in the CsA+SoC group. Both the incidence of adverse events (AE) and of the treatment-related AE were significantly higher in the CsA+SoC group (*p* 0.025 and 0.003, respectively). There were 7 SAEs, 4 of them in patients from the SoC group and 3 in the CsA+SoC group, out of which 2 in each arm were regarded as treatment related. No unexpected AEs were observed (see Table 4).

There were two readmissions during the study period that were registered as SAE (one of them also accounted for a failure to reach the primary endpoint): an episode of asthma exacerbation in a patient from the SoC group and the occurrence of acute pancreatitis in a patient from the CsA + SoC group. Both patients had a complete recovery.

Likert scale of the symptoms and well-being after discharge on the confirmed improvement in the global population, with 86.8% good–very good answers in the SoC group and 88.6% in the CsA+SoC group patients at EOS visit. In addition, X-ray improvement was stated in 35/36 of the SoC group and 43/44 of those in the CsA+SoC group.

### 3.6. Process-Related Analytes

Exploratory objectives included the time to normalization of the relevant laboratory parameters and did not yield significant differences between treatment arms. These comparisons can be found in Appendix A. Briefly, the CRP, LDH, and ferritin levels were found to decrease over the first 5 days of the trial, whereas the D-dimer levels persisted as moderately high during admission. Less movements were observed in the levels of the muscle cell markers CPK and Tnp I during the trial. Leucocyte subpopulations were determined, and, in a subgroup of patients, a full analysis of the lymphocyte subtype differentiation was included. Of note, the profound depletion of the CD4 and CD8 T cells observed in most patients on admission was recovered by Day 8, and the reconstitution of the different subpopulations was similar between both treatment arms. Lack of enough data at Days 15 and 22 hampered comparisons at these time periods. The levels of immunoglobulin classes were within a normal range at all time points and were comparable between arms. With respect to seroconversion, around half of the patients had anti SARS-CoV2 positive antibody titers at the study entry (Day 1), and the percentage increased to 96% and 81% of the IgG anti SARS-CoV2 antibodies at Day 8, respectively, in the SoC and CsA+SoC groups (Table 5).

Based on these process-related laboratory parameters, we explored independent risk factors predicting admissions lasting longer than 6 days from recruitment (post hoc analysis, Appendix A).

## 4. Discussion

We report here the negative results from a randomized controlled trial (RCT) of CsA in patients with COVID-19 pneumonia and respiratory failure on a corticosteroid background treatment. Even 3 years after the disease outbreak, no definite clues helping select candidates for immunosuppressants was found. Moreover, solid evidence supporting the role of specific immunosuppressants besides corticosteroids in improving the outcomes of patients with moderate-to-severe disease is still lacking. Emerging results from different RCT have drawn a small effect size of different strategies, a situation which underscores the need to enroll large numbers of participants. For instance, the RECOVERY Collaborative Group gathered almost 6500 patients to confirm the efficacy of dexamethasone in improving the survival of critically ill patients and of those in need of oxygen support [18]. As for anti-IL6 strategies, their efficacy in improving the principal outcomes of COVID-19 continues to be controversial according to an updated Cochrane systematic review [19], in spite of the amount of available data, while the use of other immunosuppressants cannot be recommended at this time [20,21,22].

During the first months of the pandemic, clinicians involved in COVID-19 management had the impression that these agents held higher efficacy (supported in some cases by observational study results that were to be taken with caution) than the ones found in subsequent clinical trials. This might have been due to the individualized selection criteria applied in their clinical practice. A plain explanation for the RCT failures was that COVID-19 is a complex disease where the outcome is determined by a multifactorial background, which includes not only host-intrinsic factors, such as susceptibility gene variants [23], but also those related to a saturation of health care resources, community transmission and viral load, risks associated with invasive procedures and prolonged hospitalization, the timing of referral/consultation, etc.

As concerns our study, even though controllable risk factors were equally distributed between arms, several facts may have contributed to its results, as are next discussed.

In the first place, we designed the trial in accordance with the characteristics of the patients admitted to Fundación Jiménez Díaz University Hospital during the first weeks of the pandemic. The severity of the conditions at that time was extremely high, with almost 30% of the admitted patients in need of ventilatory support. In addition, patients usually showed a flaring evolution leading to prolonged hospital stays. In those weeks, a strict lockdown protocol and the use of face masks had an immediate impact on SARS-CoV2 transmission in our environment and, most probably, on the viral load of the infected, altogether making it possible for the initial wave to subside. These circumstances completely changed the scenario for the trial. Indeed, the pandemic course in waves led to a slow and intermittent recruitment pace in different epidemiological settings, with earlier referrals and a better standard of care. The latter included the use of corticosteroids, thromboprophylaxis, and specific ventilation procedures, which together succeeded in lowering severity. We could effectively observe the impact of these measures in the interim analysis, as well as at the end of the trial, since our study population did not show the anticipated severity by a comparatively much lower mortality than the one coming from the published data, including the already mentioned RECOVERY cohort [18]. Even though we introduced an amendment to allow for the inclusion of older, more severe patients (see Appendix A), the post hoc subanalyses of age, comorbidities, or severity of respiratory failure did not help identifying the potential candidates for the use of CsA in the study population. Most probably, these subanalyses were hampered by the existence of a group of patients with mild disease, who did not deteriorate during admission and could be rapidly discharged. Notwithstanding, when we plotted the sequential FiO2 requirements, we could identify a signal for the benefit of CsA in preventing a second flare of respiratory failure in a subgroup of patients who remained admitted on Day 6 of the trial (as elaborated in Appendix A). This possibility remains highly speculative but led us to explore the risk factors for need of hospital care after 1 week of admission (as shown in Appendix A), considering that this risk could be used to identify the target population for the use of immunosuppressive therapy in future trials.

An additional point to raise is that, on the whole, our population did not reach cut-off values of hyperinflammation—except for CRP levels [2,24]—which probably defines the target population for immunosuppressant strategies. Finally, we cannot rule out that the dosing schedule of CsA was insufficient and also that the efficacy of corticosteroids could overshadow that of CsA up to a point. This fact was supported by the trends in the normalization of white cell subpopulations during the first week of the trial, which was comparable between arms.

Our study has several limitations that may hamper interpretation of the results, such as its open-label design, the need to reconfirm inclusion criteria after recruitment due to the trial characteristics, and accounting for potential randomization bias and changes in the epidemiological characteristics of the infection over time. Despite these limitations, the main strengths of this study are its pragmatic, low intervention and randomized design, with the SoC as a control group and being set up in a multicentric, national level investigation with a reasonable dropout rate.

## 5. Conclusions

In summary, our study showed that CsA did not increase the therapeutic response over the SoC in patients with COVID-19 pneumonia and respiratory failure. We suggest that the target population for this kind of strategy should be carefully selected. In addition, the scenario of inpatients with COVID-19 changed “on the go” soon after the trial started and outcome measures would have needed to be changed accordingly. We propose the use of a set of criteria predicting the risk of prolonged hospital care as a therapeutic objective in future trials in order to select a target population for immunosuppressive therapy.

## Figures and Tables

**Figure 1 jcm-13-05242-f001:**
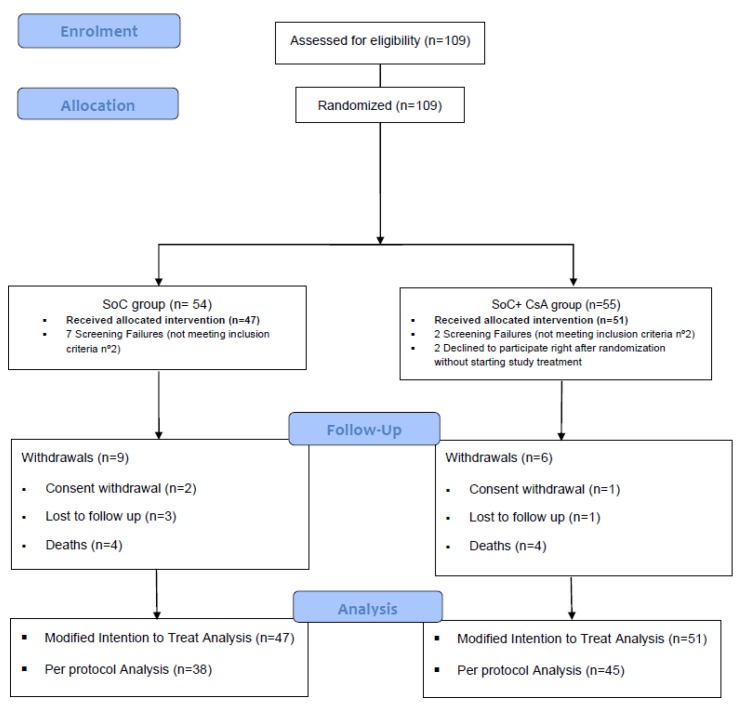
Study participant flow diagram.

**Figure 2 jcm-13-05242-f002:**
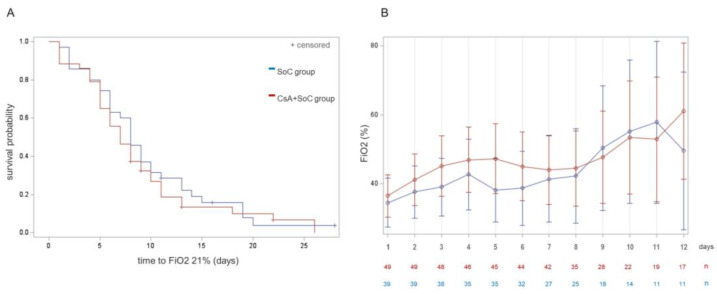
Comparison of the FiO2 requirements during hospitalization between groups. (**A**) Survival curves showing the probability of reaching FiO2 21% over time yielded no differences between the arms (log-rank test, *p* 0.48). (**B**) Daily FiO2 requirements in the ward-admitted patients (mean ± CI 95%).

**Table 1 jcm-13-05242-t001:** Participant baseline characteristics.

	CsA+SoC Group	SoC Group
Total N (women)	51	47
Female sex, n (%)	22 (43.1)	14 (29.8)
Age, mean ± SD [95% CI], median	60.2 ± 11.7 [57.0, 63.5] 60.5	62.9 ± 13.0 [59.1, 66.7] 62.0
Age ≥ 65, n (%)	21 (44.7)	21 (41.2)
Ethnicity, n (%) (Caucasian, Hispanic, and others)		
Caucasian	26 (51)	32 (68)
Hispanic	15 (29.4)	11 (23.4)
others	10 (19.6)	4 (8.6)
BMI, mean ± SD [95% CI], median	29.2 ± 5.7 [27.5, 30.9] 28.6	28.7 ± 4.1 [27.4, 30.0] 28.5
BMI ≥ 30, n (%)	16 (37.2)	14 (34.1)
Smoking habit, n (%)	(3, 13, 35)	(2, 19, 26)
Active	3 (5.9)	2 (4.3)
Past	13 (25.5)	19 (40.4)
Never smoker	35 (68.6)	26 (55.3)
^a^ Comorbidities, n (%)	24 (47.1)	29 (61.7)
Hypertension, n (%)	19 (37.3)	23 (48.9)
Diabetes, n (%)	10 (19.6)	5 (10.6)
Ischemic heart disease, n (%)	1 (2.0)	5 (10.6)
Active cancer, n (%)	3 (5.9)	1 (2.1)
Exposure to immunosuppressants, n (%)	1 (2.0)	3 (6.4)
COPD, n (%)	3 (5.9)	4 (8.5)
History of thromboembolic disease, n (%)	5 (9.8)	4 (8.5)
Time (days) from first symptom, mean ± SD [95% CI], median	8.8 ± 6.9 [6.8, 10.7] 8.0	8.5 ± 6.4 [6.7, 10.4] 8.0
Days of dyspnea, mean ± SD [95% CI], median	4.7 ± 5.2 [2.8, 6.7] 3.0	4.5 ± 3.9 [3.0, 6.0] 3.0
Severity according to CURB-65, mean ± SD [95% CI], median	0.8 ± 0.9 (0.6–1.1) 1.0	± 0.8 (0.6–1.1) 1.0

^a^ Number of participants with at least one comorbidity; BMI: body mass index; COPD: chronic obstructive pulmonary disease; CsA: cyclosporine; SoC: standard of care; CI: confidence interval; and SD: standard deviation.

**Table 2 jcm-13-05242-t002:** Primary efficacy outcomes.

% Patients without Oxygen Support at Day 12	All	CsA+SoC Group	SoC Group	Difference (95% CI)
mITT population, N	96 *	51	45	
n (%)	67 (69.8)	35 (68.6)	(71.1)	−2.5 (−20.9, 15.9, NS)
PP population, N	83	45	38	
n (%)	63 (75.9)	34 (75.6)	29 (76.3)	−0.7 (−19.14, 17.74, NS)

* A total of 2 patients withdrew consent before reaching Day 12; mITT: modified intention to treat; PP: per protocol; CsA: cyclosporine; SoC: standard of care; CI: confidence interval; and NS: not statistically significant.

**Table 3 jcm-13-05242-t003:** Secondary outcomes.

	All	CsA+SoC Group	SoC Group	*p*
Total, N	98	51	47	
All deaths, N (%)	8 (8.2)	4 (7.8)	4 (8.5)	ns
Deaths Y1 + Y2, n (%)	5 (5.6)	2 (4.3)	3 (7.0)	ns
Deaths ≥ 65 y, n (%)	7 (16.6)	3 (14.3)	4 (19.0)	ns
Discharges, n (%)	83 (84.7)	43 (84.3)	40 (85.1)	ns
LOS, mean ± SD [CI 95%], median	9.9 ± 4.8 [8.8, 10.9] 9.0	10.3 ± 4.9 [8.8, 11.8] 9.0	9.4 ± 4.8 [7.9, 10.9] 9.0	ns
LOS Y1 + Y2, mean ± SD [CI 95%], median		10.1 ± 4.9 [8.6, 11.6] 8.0	8.9 ± 4.3 [7.4, 10.5] 8.0	ns
LOS < 65 y, mean ± SD [CI 95%], median		10.4 ± 5.5 [8.2, 12.6] 8.5	8.7 ± 3.3 [7.3, 10.0] 8.5	ns
LOS > 65 y, mean ± SD [CI 95%], median		10.1 ± 4. [8.0, 12.1] 9.0	10.8 ± 6.7 [6.9, 14.7] 9.5	ns
Patients achieving FiO2 21%, n (%)	71 (72.4)	39 (76.5)	32 (68.1)	ns
LOS until FiO2 21%, mean ± SD [CI 95%], median		7.4 ± 5.4 [5.7–9.2]	7.9 ± 6.5 [6.1, 9.8]	ns
SBP at Day 16 *, mean ± SD [CI 95%], median		2.6 ± 22.5 [−18.2, 23.3] 6.0	−25.6 ± 15.2 [−44.5, −6.7] −32.0	0.036

LOS: length of stay; SBP: systolic blood pressure; ITT: intention to treat; CsA: cyclosporine; SoC: standard of care; SD: standard deviation; * difference from levels at ER; and ns: not statistically significant.

**Table 4 jcm-13-05242-t004:** Adverse events during the study.

Number of Participants with AE	Total	CsA+SoC Group	SoC Group	Comparison
All, N	98	51	47	
Related, n (%)	34 (34.7)	25 (49)	9 (19.1)	*p* 0.003
SAE, n (%)	7 (7.1)	3 (5.9)	4 (8.5)	*p* > 0.05
TOTAL, n (%)	54 (55.1)	34 (66.7)	20 (42.6)	*p* 0.025

AE: adverse event; SAE: serious adverse event; SoC: standard of care; and CsA: cyclosporine.

**Table 5 jcm-13-05242-t005:** Humoral anti-SARS-CoV2 response.

		Positive/Tested (%)			
	Isotype	Day 1	Day 8	Day 15	Day 22
CsA+SoC group	IgG	15/29 (52%)	25/31 (81%)	4/5 (80%)	0
	IgM	17/29 (59%)	24/31 (77%)	4/5 (80%)	0
SoC group	IgG	10/22 (45%)	22/23 (96%)	6/6 (100%)	1/1 (100%)
	IgM	10/22 (45%)	21/23 (91%)	4/6 (67%)	1/1 (100%)

SoC: standard of care; CsA: cyclosporine.

## Data Availability

The raw data and results from statistical analysis are available on request from the corresponding author (lucia.llanos@fjd.es). Data are not publicly available due to patient privacy issues.

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
