# Peer review of "Cyclosporin A as an Add-On Therapy to a Corticosteroid-Based Background Treatment in Patients with COVID-19: A Multicenter, Randomized Clinical Trial"

_jcm, 2024, doi:10.3390/jcm13175242_

Round 1

Reviewer 1 Report

Comments and Suggestions for Authors

The manuscript presents an evaluation of the efficacy of cyclosporine A (CsA) as an adjunctive therapy in patients with severe COVID-19 pneumonia as a randomized, multicenter trial. No differences were found compared to high-dose corticosteroid therapy. Although the study was not blinded, it was well designed and is described in detail. Patient characteristics are complete.

Comments:

Please explain the abbreviation FiO2 when first used in the article.

In Table 3, "median" is sometimes italicized and sometimes not. Please standardize.

It would be good to have a separate "Conclusions" section at the end

Author Response

Dear Reviewers and Editor,

We would like to thank you for your thoughtful revision and relevant comments made to our work,
that will undoubtfully improve its understanding. Please find below our answers point by point to the
questions raised. We have also taken the opportunity to correct some other typographic/format
errors through manuscript and supplementary files that were not previously detected. Only the
following modified files have been uploaded again:

  • Revised Manuscript with track changes and clean
    - Revised Figure 1 (already changed in the revised manuscript)
    - Simplified Supplementary file 3
    - Supplementary file 4 (typographic error)
    If you need further clarifications or comments, we will be happy to address them.

Best regards

Reviewer 1

The manuscript presents an evaluation of the efficacy of cyclosporine A (CsA) as an adjunctive
therapy in patients with severe COVID-19 pneumonia as a randomized, multicenter trial. No
differences were found compared to high-dose corticosteroid therapy. Although the study was not
blinded, it was well designed and is described in detail. Patient characteristics are complete.

Comments:

  • Please explain the abbreviation FiO2 when first used in the article: The correction can be found
    on page 3 (Fraction of Inspired oxygen)
  • In Table 3, "median" is sometimes italicized and sometimes not. Please standardize. Text format
    has been corrected. Median is now show in straight characters at Table 3.
  • It would be good to have a separate "Conclusions" section at the end. We have now re-written
    the last part of the discussion in order to include a “conclusions” paragraph and also to separately acknowledge limitations and strengths of the study (as suggested by reviewer 2).  Please find the new text at page 10 of the manuscript.

Reviewer 2 Report

Comments and Suggestions for Authors

The authors describe their randomized clinical trial, which assessed the efficacy of cyclosporin A as an add-on therapy to standard care in COVID-19 patients. They concluded that cyclosporin A did not positively impact patient outcomes in their trial. They suggest that future studies should consider outcomes that better reflect the evolving nature of the disease and focus on a more specific target population.

This manuscript presents an interesting study. Please consider the following suggestions for improving its presentation:

-          The title and abstract mention high-dose corticosteroids as the standard of care. However, the study reports that 55 patients received methylprednisolone pulses, leaving it unclear whether the remaining participants received high-dose steroids. Additionally, Supplementary File 3, which outlines other medications, is difficult to understand. Please provide more details on the standard of care practices.

-          There are discrepancies in the numbers reported in the Abstract, Results sections, and Figure 1. Specifically:

o   Abstract: “109 patients were included and 98 randomized: ...”

o   Results: “All 109 patients were randomized, but 11 of them were not considered for allocation purposes. Thus, 98 patients, 47 in the SoC group and 51 in the CsA+SoC group were considered for the ITT population.” “2 patients withdrew consent before reaching day 12, so they could not be included in ITT population for the primary efficacy outcome analysis.”

o   Figure: Three patients withdrew consent, while 98 were included in the intention-to-treat analysis.

Please consider reviewing and updating the numbers so that they match.

-          According to the statistical considerations section, randomized patients who did not receive study medications were excluded from the intention-to-treat analyses, which contradicts the standard ITT principle that all randomized patients must be included in the analyses to preserve the benefit of randomization. The authors should explain why they used a modified approach to ITT and whether this was specified in the trial protocol. The authors also stated that patients who withdrew consent were excluded from the ITT. Did these patients withdraw their consent for their data to be used as well? Please provide additional information to justify their exclusion from ITT.

-          The “Secondary Outcomes” subsection refers to groups A and B. Please clarify which study groups these are referring to.

-          Supplementary File 8 was not referenced in the manuscript.

-          Please consider citing Supplementary File 7 in the methods section as well.

-          Supplementary File 5 is cited for plotting sequential FiO2 requirements for a subgroup of patients, but this does not appear accurate. Please verify this information. Additionally, avoid introducing new results in the Discussion section (e.g., the citation to Supplementary File 6).

-          Even though the study's limitations were mentioned throughout the Discussion section, please consider dedicating paragraphs to reporting the study's strengths and weaknesses.

-          In the discussion section, the statement “All clinicians involved in COVID-19 management believe that these agents hold higher efficacy than the one found in the trials, perhaps due to the individualized selection criteria applied in their clinical practice.” Seems like an overreach. Please consider supporting it with a citation or revising the language. 

Author Response

Dear Reviewers and Editor,

We would like to thank you for your thoughtful revision and relevant comments made to our work,
that will undoubtfully improve its understanding. Please find below our answers point by point to the
questions raised. We have also taken the opportunity to correct some other typographic/format
errors through manuscript and supplementary files that were not previously detected. Only the
following modified files have been uploaded again:

  • Revised Manuscript with track changes and clean
    - Revised Figure 1 (already changed in the revised manuscript)
    - Simplified Supplementary file 3
    - Supplementary file 4 (typographic error)
    If you need further clarifications or comments, we will be happy to address them.

Best regards

Reviewer 2

The authors describe their randomized clinical trial, which assessed the efficacy of cyclosporin A as
an add-on therapy to standard care in COVID-19 patients. They concluded that cyclosporin A did not positively impact patient outcomes in their trial. They suggest that future studies should consider
outcomes that better reflect the evolving nature of the disease and focus on a more specific target
population.

This manuscript presents an interesting study. Please consider the following suggestions for
improving its presentation:

  1. The title and abstract mention high-dose corticosteroids as the standard of care. However,
    the study reports that 55 patients received methylprednisolone pulses, leaving it unclear
    whether the remaining participants received high-dose steroids. Additionally, Supplementary
    File 3, which outlines other medications, is difficult to understand. Please provide more details
    on the standard of care practices.

The study was set to reproduce clinical practice at the Quironsalud network of Public Hospitals
(QS4H) during the first wave. According to your suggestion, supplementary file 3 has been
updated to clarify the standard practice. Even though the protocol underwent some modifications
during the period of the study, it was based on a high dose corticosteroid regime, including 1-3
methylprednisolone pulses in most patients from admission. A 1-to-5-day course of 40 to 80 mg
methylprednisolone was also possible. While the local clinical investigators were encouraged to
adhere to this protocol, the final decision about dosage and regime was in their hands and could
vary in relationship to the initial response and to individual factors. In addition, since the trial was
conducted in a clinical practice setting, administration of the first dose of methylprednisolone was
possible before recruitment. In the same way, at certain point during the pandemic, patients
admitted to ICU received tocilizumab as rescue therapy followed by dexamethasone at sepsis
schedule without methylprednisolone pulses. Taken together, we believe that SoC was indeed
based on high-dose corticosteroids, moreover at a time when its use was not widespread.
Nevertheless, we acknowledge that it might an over generalization, so we have now removed
reference to “high-dose” at the title and abstract and wherever applicable in the text, while at
page 4 (results section, subheading 3.2.) we now specify dosages of intravenous
methylprednisolone received by the patients, besides pulses.

2. There are discrepancies in the numbers reported in the Abstract, Results sections, and
Figure 1. Specifically:

  • Abstract: “109 patients were included and 98 randomized: …”
  • Results: “All 109 patients were randomized, but 11 of them were not considered for
    allocation purposes. Thus, 98 patients, 47 in the SoC group and 51 in the CsA+SoC
    group were considered for the ITT population.” “2 patients withdrew consent before
    reaching day 12, so they could not be included in ITT population for the primary
    efficacy outcome analysis.”
  • Figure: Three patients withdrew consent, while 98 were included in the intention-totreat
    analysis.
  • Please consider reviewing and updating the numbers so that they match.

Figure 1 and abstract and manuscript text have been reviewed for consistency, and figure 1 has
been replaced. All 109 patients that were included were also randomized due to the study
procedures description and work routine, that stated that candidates would be selected in the emergency department or on the hospital ward, during the first 24 hours of their arrival at the
center (always as soon as possible), among those fulfilling eligibility criteria. All of these criteria
could be confirmed at inclusion except for criteria nº2 (clinical suspicion should be prospectively
confirmed with PCR or specific antibodies). Thus, there were 9 patients (7 randomized to the
control group and 2 randomized to the experimental group) in whom clinically suspected
COVID19 pneumonia could not be confirmed in the following days. We decided not to include
these patients in the ITT population because we considered they were screening failures
(although randomized, they did not have COVID19 in the end). Also, 2 patients randomized to
experimental group withdrew consent right after randomization but before receiving study drug.
Finally, 98 patients were considered for the modified ITT population. Primary efficacy outcome
could only be assessed in 96 out of 98, since 2 of the 3 patients that withdrew consent did it
before reaching this time point (12 days). Notwithstanding, these 2 patients were daily evaluated
for the other variables of the study both as regards efficacy and safety until withdrawal.

During study treatment and follow up period, there were 15 patient withdrawals, 9 in the SoC
group and 6 in the Csa+SoC group, making finally 83 patients for the per protocol population.

3. According to the statistical considerations section, randomized patients who did not receive
study medications were excluded from the intention-to-treat analyses, which contradicts the
standard ITT principle that all randomized patients must be included in the analyses to
preserve the benefit of randomization. The authors should explain why they used a modified
approach to ITT and whether this was specified in the trial protocol.

A modified approach to ITT analysis (mITT) was foreseen in the study protocol (see supplementary
file 8, in Spanish), that stated: For both efficacy and safety assessment, an intention-to-treat analysis
will be conducted, including all randomized patients who have received at least one dose of the study
medication, and a per-protocol analysis, which shall include patients who have completed the study.

An early recruitment and randomization were considered critical for the development of the study,
which was on the other hand set as a pragmatical clinical trial. As such, some biases were accepted,
such as those dependent on open label or individual adjustments of corticosteroids and adjuvant
treatments. Please note that we now include those limitations at the discussion of the manuscript.
We have also made the correction of ITT to modified ITT (mITT) as indicated.

In the case of our study, with an open label design, patients in whom participation could have been
influenced by group allocation were those that withdrew consent after randomization. As shown in
figure 1, in total there were 2 consent withdrawals in SoC and 3 in CsA +SoC group (2 of them before
receiving study medication). The rest of patients excluded in the modified ITT analysis were related
to screening failures.

The authors also stated that patients who withdrew consent were excluded from the ITT. Did these
patients withdraw their consent for their data to be used as well? Please provide additional
information to justify their exclusion from ITT.

As regards to data use for mITT purposes, according to our ICF for the trial, no further data could be
collected after consent withdrawal. Therefore, the primary response could not possibly be assessed
in those subjects who declined participation before day 12 (2 of them withdrew after randomization
but before receiving study medication, and thus were not included in de mITT population, and 2 more
patients that withdrew after receiving study medication but before day 12, and thus were included in
the mITT population but not in the primary outcome analysis).

4. The “Secondary Outcomes” subsection refers to groups A and B. Please clarify which study
groups these are referring to.

The correction has been done in Section 3.3.

5. Supplementary File 8 was not referenced in the manuscript. Please consider citing
Supplementary File 7 in the methods section as well.

A reference to supplementary file 8 (study protocol) and 7 (protocol amendments) has been included
in section 2.1.

6. Supplementary File 5 is cited for plotting sequential FiO2 requirements for a subgroup of
patients, but this does not appear accurate. Please verify this information. Additionally, avoid
introducing new results in the Discussion section (e.g., the citation to Supplementary File 6).

With regards to the first part of the question, reference has been reviewed and corrected to
supplementary file 4, as it was a mistake.

With regards to the second part of the question, we have now included a brief reference to
Supplementary file 6 at the results section, page 7 and also re-written the comment at the discussion
section.

7. Even though the study's limitations were mentioned throughout the Discussion section,
please consider dedicating paragraphs to reporting the study's strengths and weaknesses.

The discussion section has been reviewed to reorganize and add some information about study
strengths and weaknesses.

8. In the discussion section, the statement “All clinicians involved in COVID-19 management
believe that these agents hold higher efficacy than the one found in the trials, perhaps due
to the individualized selection criteria applied in their clinical practice.” Seems like an
overreach. Please consider supporting it with a citation or revising the language.

The statement has been reviewed and modified in order to better express the idea and possible
reasons that the clinical impression of efficacy of various of these agents has not been subsequently
supported by RCT results.
